# Electronic correlations and transport in iron at Earth's core conditions

L. V. Pourovskii [1,2✉], J. Mravlje[3], M. Pozzo [4] & D. Alfè [4,5]

The transport properties of iron under Earth's inner core conditions are essential input for the geophysical modelling but are poorly constrained experimentally. Here we show that the thermal and electrical conductivity of iron at those conditions remains high even if the electron-electron-scattering (EES) is properly taken into account. This result is obtained by ab initio simulations taking into account consistently both thermal disorder and electronic correlations. Thermal disorder suppresses the non-Fermi-liquid behavior of the body-centered cubic iron phase, hence, reducing the EES; the total calculated thermal conductivity of this phase is 220 $Wm^{-1}K^{-1}$ with the EES reduction not exceeding 20%. The EES and electron-lattice scattering are intertwined resulting in breaking of the Matthiessen's rule with increasing EES. In the hexagonal close-packed iron the EES is also not increased by thermal disorder and remains weak. Our main finding thus holds for the both likely iron phases in the inner core.

[1] CPHT, CNRS, Ecole Polytechnique, Institut Polytechnique de Paris, Route de Saclay, 91128 Palaiseau, France. [2] Collège de France, 11 place Marcelin Berthelot, 75005 Paris, France. [3] Jozef Stefan Institute, SI-1000 Ljubljana, Slovenia. [4] Department of Earth Sciences and London Centre for Nanotechnology, University College London, Gower Street, London WC1E 6BT, UK. [5] Dipartimento di Fisica Ettore Pancini, Università di Napoli Federico II, Monte S. Angelo, I-80126 Napoli, Italy. ✉email: leonid@cpht.polytechnique.fr

Knowing the transport properties of iron at high-pressure and high-temperature conditions relevant to the Earth's core is essential for modeling the Earth's geodynamo mechanism that maintains the Earth's magnetic field[1]. The key quantity is the thermal conductivity that needs to be low enough for the transfer of heat in the liquid outer core to proceed by a convective mechanism[2]. Whereas quite low values of thermal conductivity were assumed (based on extrapolation of low-temperature data to higher temperatures) for a long time[3,4], recent first-principle calculations[5,6] reported higher values.

These results have important implications. If the thermal conductivity is high, convection can be sustained only by a large heat flow[4] associated with a younger Earth's inner core (some evidence for the latter was recently reported[7]). The thermal conductivity of solid iron is a key input for models of the inner core's anisotropy[8–10]; a rather high conductivity has been recently predicted for hexagonal close-packed (hcp) $\epsilon$-Fe at the relevant conditions[11,12]. Experimentally, reaching high temperatures and high pressures is challenging, and there is large scatter in the data from direct measurements[13–15]. The transport properties are not the only ones to be determined with poor confidence: whereas the main constituent of the solid inner core is usually assumed to be the hcp $\epsilon$ phase, some experiments show evidence of the body-centered cubic (bcc) phase[16,17]. bcc-Fe was also predicted to be the stable phase of iron at inner-core conditions by recent molecular dynamics simulations[18].

From the theoretical side, the transport mechanisms in iron at Earth's core conditions are also incompletely understood with several important fundamental questions not resolved to date: (i) whereas and to what extent the electronic correlations affect the transport properties is being intensely debated[12,19–21], (ii) the question of the interplay of the EES and electron-lattice scattering (ELS), namely: Is the Mathiessen's rule that estimates the total scattering rate from the sum of the individual ones valid? (iii) Does the Wiedemann–Franz law that relates the thermal and electrical conductivity apply? For correlated metals in a Fermi liquid regime, the proportionality constant (the Lorenz number) for the EES is greatly affected by the energy dependence of the corresponding inelastic scattering rate[12].

For crystalline hcp-Fe, Pourovskii et al.[12] found a moderate impact of electronic correlations with the EES being far less important than the previously calculated[11] ELS. They speculated that the EES could increase if the large thermal disorder due to the motion of the ions close to (and above) the melting temperature was taken into account. Xu et al.[20] evaluated EES for the perfect hcp lattice and combined it with a separately calculated ELS term using Mathiessen's rule. They obtained a significant reduction of the total conductivity, and found the EES further increasing when thermal disorder in the liquid state was included.

The questions on the relative impact of EES and its interplay with the ELS are also highly relevant if the main constituent of the inner core is the bcc iron phase[16–18]. The bcc phase has been pointed out to be significantly more correlated, compared with hcp-Fe, due to a van-Hove singularity in its electronic structure slowing down electrons[19,22,23]. Dense bcc iron is expected to exist only at high temperatures close to melting; its stabilization is predicted to be induced by anharmonic ionic vibrations[24] or a complex self-diffusion mechanism[18]. Very strong deviations from the perfect bcc crystalline order are thus expected. Therefore, possible impact of these distortions on electronic correlations needs to be taken into account. Intuitively, one might expect the crystalline disorder to slow down the electrons further, which would enhance the effects of electronic correlations. The impact of intertwined crystalline disorder and many-electron effects on transport has not been assessed in previous theoretical studies in iron at Earth's core conditions[12,25], except by Hausoel et al.[19] who

evaluated the strength of correlations in a thermally disordered face-centered cubic (fcc) phase of Ni at Earth's core conditions finding the impact of lattice vibrations on correlations insignificant, and Xu et al.[20] who, conversely, found that the thermal disorder (whose effects they evaluated in the liquid state) increases the EES significantly.

In the present work, we investigate these questions by considering electron transport in iron under conditions relevant to the inner core. Iron at these conditions is modeled by the density-functional + dynamical mean-field theory (DFT+DMFT) method[26–28] applied to a set of Fe supercells (SCs) randomly chosen from configurations produced by molecular dynamics (MD) simulations, as done earlier in refs. [19,20]. Our transport calculations thus include both the effects of lattice distortions due to the thermal motion of ions and the electronic correlations. Focusing on the bcc phase, where both the electronic correlations and complex nonharmonic lattice degrees of freedom are expected to play an important role[18,22–24,29], we find that the positional disorder does affect the electronic correlations. However, in contrast to expectations, we find that their strength is suppressed. The van Hove singularity present in perfect bcc-Fe[23,30] is smoothed by thermal disorder, which thus reduces the EES. The total thermal conductivity is to a large extent determined by the effects of thermal disorder alone, and reduced by less than one-quarter by the EES. Even with the EES artificially increased above our calculated value, its impact on the total resistivity is far less than expected on the basis of Matthiessen's rule. We derive a qualitative explanation of this surprising result that holds universally in the limit of strong disorder, i.e., in proximity to melting temperatures (and above). Since the thermal disorder wipes out the sharp characteristic features of the DOS, one may expect a similar behavior of the EES for hcp, fcc, and liquid iron. We verify the generality of our conclusions with explicit DFT+DMFT calculations of transport in the thermally disordered hcp phase, for which we obtain values of the total conductivity and EES contribution that are very similar to those for bcc. The relative insensitivity of the transport properties to both increase of the EES and particularities of the lattice structure thus implies a robustly high conductivity in the presence of a strong EES also in other solid and liquid iron phases.

## Results

**Electron–electron scattering in the presence of thermal disorder.** We first present our key results on the conductivity of bcc-Fe at the inner core conditions. The resulting total electrical resistivity and thermal conductivity at temperature $T = 5802$ K for our set of randomly selected $3 \times 3 \times 3$ SCs are shown in Fig. 1 (open triangles; the average over the SCs is shown with the bold star) together with the results for the perfect bcc and hcp[12] lattices (for which the temperature dependence is also shown). The resistivity of perfect lattices is due to the EES term only; the temperature dependence in the case of bcc exhibits a rather slow non-Fermi-liquid[23] increase in contrast to the Fermi-liquid hcp phase[12]. The ELS starts contributing once lattice distortions are included. With the magnitude of the distortions predicted by our MD simulations, the ELS is by far the dominating term, as one sees by comparing the total conductivities with purely ELS ones also shown in Fig. 1.

One may notice that the spread of calculated conductivities within the set is quite small. The value of the total bcc-Fe thermal conductivity averaged over all 8 distorted SCs in our set is equal to 220 W m$^{-1}$ K$^{-1}$ compared with 584 W m$^{-1}$ K$^{-1}$ for the EES only (perfect bcc lattice) and 275 W m$^{-1}$ K$^{-1}$ for the ELS only; the latter is evaluated within DFT[31,32], see "Methods". For the electrical resistivity, the corresponding values are $6.02 \times 10^{-5}$,

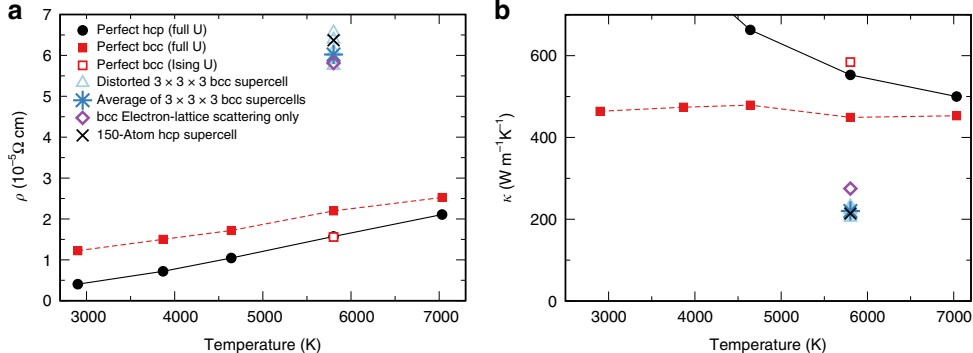

**Fig. 1 Electrical and thermal conductivity of iron under Earth's inner-core conditions. a** The total resistivity of iron at Earth's core volume (7 Å³ per atom, density 13.2 g cm⁻³) obtained by the density-functional theory + dynamical mean-field theory (DFT+DMFT) method. The curves vs. temperature with filled circles/squares are for the perfect hexagonal close-packed (hcp)[12] and body-centered cubic (bcc) unit cells, respectively, calculated with the full rotationally invariant Coulomb vertex **U**. The rest of shown data is obtained using the density–density (Ising) vertex. The empty red square is the perfect bcc result. In the case of perfect lattices, the electron-lattice-scattering (ELS) term is absent, and the resistivity is due to the electron–electron-scattering (EES) term only. The empty triangles are for eight different distorted $3 \times 3 \times 3$ bcc supercells. The bold blue star is the average value over the distorted bcc supercells. The black cross is the average value over the representative 150-atom hcp supercell. The ELS resistivity calculated within DFT for bcc (purple empty diamond) is also shown. **b** The total thermal conductivity of iron at Earth's core volume obtained by DFT+DMFT. The meaning of symbols is the same as for panel (**a**).

$1.56 \times 10^{-5}$, and $5.82 \times 10^{-5}$ Ω·cm, respectively. By comparing the total conductivity $\kappa$ with electron-lattice $\kappa_{el-lat}$, one infers that the EES reduces the thermal conductivity by about 20%; its impact on electrical resistivity is even smaller. It is instructive to check the validity of the Wiedemann–Franz law $\kappa = L_0 \sigma T$, where $L_0 = 2.44 \times 10^{-8}$ WΩK⁻² is the standard Lorenz number, by taking the ratios of thermal and electrical conductivity. For the perfect lattice, we obtain $L = 1.57 \times 10^{-8}$ WΩK⁻², whereas for the distorted one, we obtain $L = 2.28 \times 10^{-8}$ WΩK⁻². The former result deviates from the standard Lorenz number due to the frequency-dependent inelastic scattering rate. The latter result is close to $L_0$. As pointed out above, thermal disorder provides the largest contribution to the total scattering. This dominant contribution of thermal disorder, in contrast to the inelastic EES, influences the thermal and electrical conductivity equally resulting in an almost standard value for the Lorenz number.

**Self-energy and spectral function.** The electronic correlations, which are the origin of EES, are described in our framework by the local electronic self-energy. We have computed this quantity for a set of eight $2 \times 2 \times 2$ bcc SCs with the fully self-consistent DFT+DMFT method (see Methods section). In order to elucidate the effect of thermal disorder on electronic correlations, one may compare the self-energy calculated in distorted SC with that in perfect lattices. Of particular relevance to transport is the low-frequency behavior of the imaginary part of the Matsubara[33] self-energy $Im\Sigma(i\omega_n)$. The extrapolation of $|Im\Sigma|$ to low frequencies characterizes the electron-scattering rate, and the slope of the approach characterizes the electronic renormalization $m^*/m = [1 - dIm\Sigma(i\omega_n)/di\omega_n]|_{\omega_n \to 0}$. Larger magnitude of $Im\Sigma(i\omega_n)$ thus points to stronger correlations. The calculated imaginary part of the Matsubara self-energies for different Fe 3$d$ orbitals and atoms in the set of SCs is shown in Fig. 2a, together with those for the perfect bcc and hcp lattices.

One sees that thermal disorder significantly modifies the electronic self-energies. The self-energies at different positions differ, and the largest differences are comparable to the value itself. One also sees that, unlike in the case of the perfect bcc structure, where there is a clear distinction between the more correlated non-Fermi liquid $e_g$ and the less correlated Fermi-liquid $t_{2g}$ self-energy, the self-energies of different atoms and orbitals quasi-uniformly span the full range of values. Therefore, thermal disorder is

sufficiently large for the resulting self-energies not to resemble those of the perfect bcc structure. (If the disorder were smaller, one would still resolve the $t_{2g}/e_g$ blocks). The self-energy $\langle\Sigma\rangle$ averaged over all sites and orbitals of all eight SCs is close to the bcc result for the less correlated $t_{2g}$ orbital and to the Fermi-liquid self-energies of hcp-Fe. This average self-energy, analytically continued to the real axis, Fig. 2b, was used to evaluate the conductivity in the $3 \times 3 \times 3$ SCs. The more correlated bcc $e_g$ result represents a rough upper bound. The average scattering rate (given by $-\langle Im\Sigma(\omega)\rangle$) vs. $\omega$ exhibits a characteristic Fermi-liquid parabolic shape (Fig. 2b), with its value at the Fermi level, $-\langle Im\Sigma(\omega = 0)\rangle = 84$ meV, being close to the value of 90 meV previously obtained for hcp-Fe[12]. The thermal disorder thus reduces the electronic correlations compared with the perfectly crystalline bcc result.

In order to understand the origin of this effect, it is convenient also to look at the corresponding DFT+DMFT spectral function. Namely, the stronger correlations for the $e_g$ orbital in the case of bcc structure occur due to the proximity to a van-Hove singularity[19,22,23,34]. The 3$d$ spectral function averaged over all sites in all eight $2 \times 2 \times 2$ SCs is shown in Fig. 2c and is compared to the one for the perfect bcc. One sees that the narrow peak of $e_g$ states in the vicinity of $\omega = 0$ is almost completely smeared by lattice distortions. This smearing of the low-frequency peak in turn leads to weaker correlations. One thus has a counterintuitive situation that the disorder actually leads to a suppression of electronic scattering at low energies.

**Evolution of transport as a function of distortion.** It is important to notice that whereas the electronic self-energies are actually suppressed when evaluated in the presence of thermal disorder, as shown above, the calculated resistivity is strongly increased. This implies that the increase must be attributed to the ELS due to the distortions.

In order to understand better the influence of distortions, it is convenient to study the evolution of transport vs. distortion strength. To that end, we chose a representative $2 \times 2 \times 2$ SC, for which the calculated thermal conductivity of 240 Wm⁻¹ K⁻¹ is close to the one obtained from the average over the whole $2 \times 2 \times 2$ set. The atomic coordinate $\mathbf{R}_i$ of a given site $i$ in this $2 \times 2 \times 2$ SC can be written as $\mathbf{R}_i = \mathbf{R}_i^0 + \boldsymbol{\tau}_i$, where $\mathbf{R}_i^0$ denotes the position of the atom in the perfect $2 \times 2 \times 2$ SC and $\boldsymbol{\tau}_i$ is the corresponding

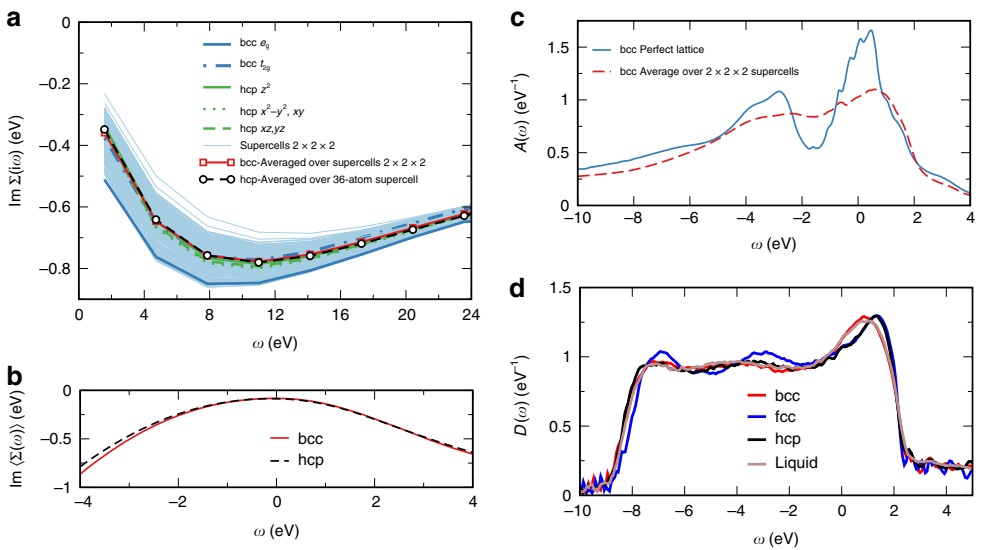

**Fig. 2 Electronic self-energy and electronic structure of iron at Earth's inner-core conditions. a** The imaginary part of dynamical mean-field-theory (DMFT) self-energy on the Matusbara grid for different atomic sites and orbitals of the set of 2 × 2 × 2 body-centered cubic (bcc) supercells (thin light-blue curves). The average over all sites and orbitals of bcc supercell self-energy $\langle\Sigma\rangle$ (thick red line with squares) and the average self-energy over sites and orbitals of the hexagonal close-packed (hcp) 36-atom supercells (black line with circles) are also shown. The blue and green curves are the self-energies for the nondegenerate orbitals in the perfect bcc and hcp lattices, respectively. **b** The average DMFT bcc (red) and hcp (dashed black) self-energy in real frequency. **c** The DMFT spectral function for the Fe $3d$ states averaged over all sites in the set of 2 × 2 × 2 bcc supercells (red dashed line) compared to that for the perfect bcc lattice (solid blue line). **d** The density-functional-theory (DFT) densities of states of the bcc, face-centered cubic, hcp, and liquid iron at Earth's core density 7 Å$^3$ per atom and $T = 6000$ K obtained by DFT molecular dynamics simulations.

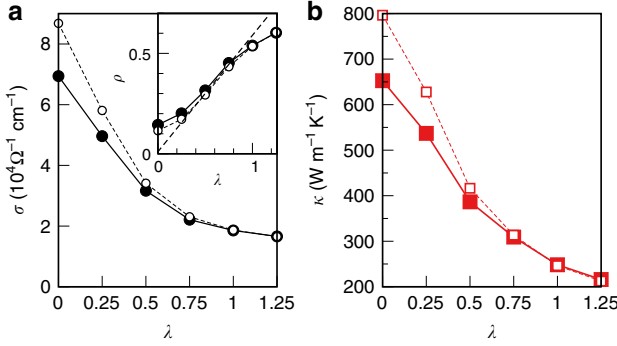

**Fig. 3 Total conductivity vs. the degree of lattice distortion.** The electrical (**a**) and thermal (**b**) conductivities are displayed as a function of the distortion level $\lambda$ varying from 0 (perfect lattice) to 1 (full distortion), and beyond. The solid lines (full symbols) are calculated with full self-consistent density-functional theory + dynamical mean-field theory (DFT+DMFT) method at the corresponding distortion levels; dashed lines (empty symbols) are obtained using the averaged self-energy $\langle\Sigma\rangle$ at full distortion $\lambda = 1$. The inset in (**a**) shows the corresponding resisitivity $\rho$ and indicates the onset of saturation at large $\lambda$ with deviation from linearity (straight dashed line).

displacement in the distorted cell. Introducing a scaling parameter $\lambda$ for the distortion, $\mathbf{R}_i^\lambda = \mathbf{R}_i^0 + \lambda\boldsymbol{\tau}_i$, one can smoothly tune the distortion from the vanishing one $\lambda = 0$, to the actual distortion in the molecular dynamics simulation snapshot, $\lambda = 1$, and beyond.

We performed full self-consistent DFT+DMFT calculations for a set of $\lambda$; the resulting electrical and thermal conductivities vs. $\lambda$ are shown in Fig. 3. One can identify the regime of weak disorder with, in particular, a rather rapid decay of $\kappa$ vs. $\lambda < 0.25$. By inserting the average self-energy $\langle\Sigma\rangle$ of fully distorted SCs instead of the DMFT self-energies calculated at a given $\lambda$, one obtains a significantly larger conductivity at $\lambda = 0$ and a yet steeper decay. The difference in behavior of conductivities vs.

$\lambda < 0.25$ for those two cases is due to the evolution of electronic self-energies upon increasing distortions. Namely, the distortions suppress the non-Fermi-liquid behavior of $e_g$ orbitals (see Fig. 2) and reduce the overall magnitude of scattering $\sim |\mathrm{Im}\Sigma(\omega)|$; this reduction of EES partially compensates the enhancement of ELS with $\lambda$.

The decrease in the conductivity becomes drastically slower for $\lambda > 0.5$. This saturation of conductivity clearly seen in Fig. 3 is due to the phenomenon of resistivity saturation that is known to occur in weakly correlated metals at elevated temperatures[35], and to play an important role for the transport in Fe[6]. Namely, in weakly correlated metals, the resistivity saturates at a so-called Mott–Ioffe–Regel value that corresponds to a scattering length equal to the minimal interatomic distance. As one sees in the inset of Fig. 3a, the electrical resistivity vs. $\lambda$ starts deviating from the linear scaling for $\lambda \geq 0.75$, thus exhibiting this saturation effect. Notice that a change in $\lambda$ emulates a change in temperature, as phonon displacement is proportional to temperature. Hence, at our simulation temperature of 5802 K, corresponding to $\lambda = 1$, the onset of saturation already occurred.

On the other hand, when electronic correlations become dominant, the resistivity can grow further, which is called the bad-metal regime[35]. This is evidenced by the data in Fig. 4b, where we artificially increased the strength of the ESS, as discussed below. We stress that our approach correctly reproduces both the phenomenon of resistivity saturation and the bad-metal regime, whichever is applicable in the given case.

We also notice that for $\lambda > 0.5$, the evolution of transport vs. $\lambda$ obtained with full DFT+DMFT calculations becomes basically indistinguishable from that obtained with the site and orbital average self-energy $\langle\Sigma\rangle$ at the real distortion, $\lambda = 1$. This implies that the evolution of self-energy vs. $\lambda$ has no impact on the transport at large distortions $\lambda > 0.5$. Moreover, in this strong disorder regime, the site and orbital fluctuations of the self-energy do not affect the conductivity. Based on these two observations,

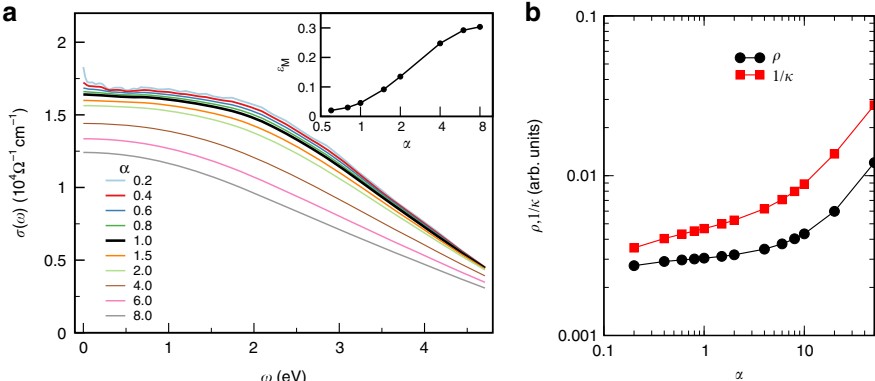

**Fig. 4 Electron–electron scattering and Matthiessen's rule in body-centered cubic Fe. a** The optical conductivity of a representative body-centered cubic $3 \times 3 \times 3$ supercell vs. scale factor $\alpha$ applied to the imaginary part of dynamical mean-field-theory (DMFT) self-energy. The inset shows the relative deviation of calculated thermal conductivity from the Matthiessen's rule as a function of $\alpha$. **b** Calculated thermal and electrical resistivity as a function of the scale factor $\alpha$. Notice the slow dependence in the vicinity of $\alpha = 1$.

one can evaluate the transport for large supercells in the strong-disorder regime even without explicit knowledge of self-energy at every site, but simply resorting to $\langle \Sigma \rangle$ obtained with DFT+DMFT calculations for smaller cells at a similar level of thermal disorder.

**Optical conductivity and interpretation of the results**. To investigate the interplay between thermal disorder and the EES further, we also artificially tuned the strength of EES by multiplying the imaginary part of DMFT self-energy by a constant $\alpha$, and evaluated the transport for a fixed distorted lattice. Namely, we calculated the optical conductivity Eq. (8) as a function of the scale factor $\alpha$, i.e., with the self-energy $\Sigma_\alpha(\omega) = \text{Re}\langle \Sigma(\omega) \rangle + \alpha \text{Im}\langle \Sigma(\omega) \rangle$ used for all orbitals and sites of a SC. Such a "cartoon"-scaled self-energy, though not satisfying the exact Kramers–Kronig relations between $\text{Re}\Sigma(\omega)$ and $\text{Im}\Sigma(\omega)$, does capture the main features of the strong EES regime in "Hund's system" like iron: the states away from the Fermi surface have short lifetimes due to correlations, but do not have strongly reduced dispersions[36]. Uniformly scaling both the real and imaginary parts would respect the Kramers–Kronig relations, but in result, one would apply a large coherent Fermi-liquid renormalization to $3d$ bands far from the Fermi surface. The latter is a physically incorrect representation of the strongly correlated limit in iron.

A larger SC size is preferable in calculations with the scaled self-energy $\Sigma_\alpha$ in order to properly evaluate the DC-conductivity limit at small values of $\alpha$. We thus chose a representative $3 \times 3 \times 3$ SC with the conductivities at $\alpha = 1$ very close to the average one (see Fig. 1). The resulting optical conductivity vs. $\alpha$ is displayed in Fig. 4.

The calculated optical conductivity is generally smooth and virtually constant at small $\omega \to 0$ due to a strong disorder suppressing the Drude peak coming from the intraband transitions in favor of low-lying interband transitions. At small $\alpha \leq 0.4$, the Drude peak is still present, due to insufficiency of the $3 \times 3 \times 3$ SC size to describe the electron-lattice scattering for very long-lived quasiparticle states. One may notice a rather slow decrease in conductivity vs. $\alpha$; the DC conductivity $\sigma(\omega = 0)$ is reduced by less than half of its initial value with the EES increased by a factor of 40.

Matthiessen's rule predicts the total (electrical or thermal) resistivity to be given by a sum of contributions due to separate scattering mechanisms. In the present case, the total Matthiessen's rule thermal conductivity $\kappa_M$ is given by $1/\kappa_M = 1/\kappa_{el-lat} + 1/\kappa_{el-el}$, where our calculated $\kappa_{el-lat} = 275 \, \text{Wm}^{-1} \text{K}^{-1}$ and $\kappa_{el-el}$ for given $\alpha$ is calculated in the perfect bcc lattice from $\Sigma_\alpha(\omega)$. The

resulting relative deviation $\epsilon_M = \frac{\kappa - \kappa_M}{\kappa_M}$ of Matthiessen's rule conductivity with respect to the actual one, i.e., the one calculated directly in the representative SC from $\Sigma_\alpha(\omega)$, is plotted in the inset of Fig. 4a as a function of $\alpha$. One sees that Matthiessen's rule is satisfied reasonably well at a small EES, but at the large EES limit, the deviation becomes significant with Matthiessen's rule under-estimating the conductivity by about one-third. For the largest considered value of $\alpha = 8$, for which $\kappa_{el-el} = 148 \, \text{Wm}^{-1} \text{K}^{-1}$, Matthiessen's rule $\kappa_M = 96 \, \text{Wm}^{-1} \text{K}^{-1}$ is markedly smaller than the actual total conductivity $\kappa = 125 \, \text{Wm}^{-1} \text{K}^{-1}$.

In order to understand this effect, we return to the expression for the DMFT conductivity (Eq. (8)). To simplify the discussion, it is convenient to neglect site/orbital dependence of the self-energy (which we argued above does not play an important role at the relevant conditions). Under this assumption, the self-energy has no momentum dependence, and the DC conductivity can be written as

$$\sigma \propto \int d\omega \sum_k \sum_{\nu\nu'} |v_{k\nu\nu'}|^2 A_{k\nu} A_{k\nu'} (-df/d\omega). \quad (1)$$

The spectral function $A_{k\nu} = -\frac{1}{\pi} \text{Im}(\omega + \mu - \epsilon_{k\nu} - \Sigma)^{-1}$ can be further simplified by assuming that the self-energy is well described by a constant $\Sigma \sim -i\Delta$. The spectral functions hence take a Lorentzian form with the width given by $\Delta$. We observed that large disorder leads to a behavior that enables a further simplification of the expression above: (i) The eigenenergies become approximately equidistantly distributed, $\epsilon_{k\nu+1} - \epsilon_{k\nu} \approx C$, where $C$ is a constant. (ii) The current matrix element $|v_{k\nu\nu'}|$ loses strong dependence on $\nu$, $\nu'$ and essential physics can be reproduced by substituting it by a constant. Under these assumptions, the conductivity simplifies to a form

$$\sigma \propto \int d\omega v^2 \sum_k \sum_\nu A_{k\nu} \sum_{\nu'} A_{k\nu'} (-df/d\omega)$$
$$= v^2 \int d\omega \left( \int d\epsilon A_\epsilon(\omega) \right)^2 (-df/d\omega), \quad (2)$$

where in the last equality, we replaced the summation over band energies by a Riemann integral

$$\sum_\nu A_{k\nu} \to \frac{1}{\pi C} \int_{-W/2}^{W/2} d\epsilon \frac{\Delta}{(\omega + \mu - \epsilon)^2 + \Delta^2}, \quad (3)$$

where $W$ is the bandwidth.

Now, when $W \gg \Delta$, the Riemann integral reduces to integration of a Lorentzian over range $(-\infty, \infty)$ and can be approximated by a

constant. The resulting $\sigma$ hence does not depend on $\Delta$ anymore. Only when one increases the electronic scattering above the full bandwidth ($\Delta \gg W$), a new regime occurs, where integral (3) becomes $\propto W/\Delta$. In this regime, the resistivity strongly ($\propto \Delta^2$) increases as a function of $\Delta$. One sees a crossover to this regime in Fig. 4b at very large $\alpha$. However, in the relevant range $\alpha \sim 1$ enhancement of $\Delta$, reducing the contribution of each individual $A_{k\nu}$, at the same time leads to the spectral functions further from Fermi energy contributing to the DC conductivity. This increase in the number of contributing $\nu$ "conduction channels" hence compensates for the loss of conductivity due to a shorter electronic lifetime. In result, the dependence of $\sigma$ on $\Delta$ at $\alpha \sim 1$ is weak as seen in Fig. 4b. That is in contrast to the usual Drude behavior $\sigma \propto 1/\Delta$ occurring when the EES contribution (subsequently combined with ELS using Matthiessen's rule) is calculated for the undistorted lattice.

**Electronic structure and conductivity of the hcp-Fe phase.** What are the implications of our analysis for the case of the hcp phase, which is another candidate for the inner core's principal component, and for the outer core liquid phase? Should one expect a different behavior with stronger influence of EES from that found for bcc-Fe? We show below that in fact this is not the case. The EES magnitude comes out to be very similar. In fact, the distinction between crystalline phases (as well as between them and the liquid phase) becomes smaller with increasing motion amplitude of the ions, and peculiar features of their electronic structure are washed away. We compare the corresponding DFT densities of states obtained by averaging over MD configurations in Fig. 2d. Indeed, the difference between various iron crystalline phases as well as between them and liquid is quite insignificant. In contrast, it is striking in the case of the corresponding perfect lattices at Earth's core density. Namely, the DFT density of states of perfect bcc-Fe (and its DFT+DMFT spectral function, Fig. 2c) features a peak at the Fermi level $E_F$ due to the van-Hove singularity, while the density of states of perfect hcp-Fe features a dip at $E_F$ (see, e.g., Fig. 3 of ref. [23]).

We verified the validity of these qualitative arguments by explicit calculations of the conductivity for a representative configuration of thermally disordered hcp-Fe. We employed the same DFT+DMFT approach as in the bcc case, see Methods for more details of these calculations. The electronic self-energy in the thermally disordered hcp phase is found to preserve its Fermi-liquid character[12]; the average hcp self-energy is virtually coinciding with the bcc one, see Figs. 2a and b. The resulting values of total (ELS+EES) electrical resistivity and thermal conductivity for hcp-Fe are $6.37 \times 10^{-5}$ $\Omega \cdot$cm and 214 Wm$^{-1}$ K$^{-1}$, respectively. The reduction due to the EES is 9 and 24% for the electrical and thermal conductivity, respectively. As one sees (Fig. 1), the thermal conductivity of thermally disordered hcp-Fe is virtually the same as that of thermally disordered bcc. Our value for the total (ELS+EES) thermal conductivity of hcp-Fe, 214 Wm$^{-1}$ K$^{-1}$, is higher than the previous theoretical values of 190 and 147 Wm$^{-1}$ K$^{-1}$ reported by Pourovskii et al.[12] and Xu et al.[20], respectively. Both refs. [12] and [20] employed Matthiessen's rule to evaluate the total thermal conductivity; as shown above, this rule underestimates the conductivity. By applying Matthiessen's rule to combine our ELS contribution of 280 Wm$^{-1}$ K$^{-1}$ with the EES one of 605 Wm$^{-1}$ K$^{-1}$, the latter is calculated in the perfect hcp lattice using the corresponding average self-energy (Fig. 2b); we obtain the same total conductivity as ref. [12]. Matthiessen's rule thus leads to an underestimation of the hcp conductivity by about 10%. Besides resorting to Mathiessen's rule, Xu et al.[20] employed a less elaborate evaluation of the ELS conductivity, in which the saturation effect was imposed through a parallel resistor correction. Their method gives a markedly smaller ELS conductivity in the relevant regime as compared with our MD-based approach that naturally includes the resistivity saturation, see Fig. 3.

## Discussion

In conclusion, our main results are: (1) the overall reduction of iron conductivity due to the EES is only about 20%; this reduction is too weak to significantly alter the picture of a highly conductive core matter previously obtained by the DFT first-principles calculations[5,6,11]. The resulting total thermal conductivity is above 200 Wm$^{-1}$ K$^{-1}$ for both the likely solid iron phases in the inner core, bcc and hcp. Nevertheless, the EES reduction is not negligible and should be, in general, included in the detailed modeling of the inner core, (2) in bcc-Fe, the EES is not increased but rather suppressed by thermal disorder, and (3) in the relevant regime, the total conductivity exhibits markedly weaker dependence on the EES as compared with predictions of the simple Matthiessen's rule.

With drastic differences in the electronic structure of different iron phases being suppressed by thermal disorder, a similar behavior can also be expected for the fcc and liquid iron phases at the core conditions. Intuitively, substitutional disorder and small-atom impurities are not likely to have an important effect in the presence of strong thermal disorder; however, the impact of alloying with light elements on the EES needs to be evaluated by direct calculations. Finally, given a slow dependence of the total thermal conductivity on the magnitude of the EES in the thermal-disorder-dominated regime demonstrated in Fig. 4, it is unlikely that electronic correlations would drastically affect the transport properties, even if the magnitude of EES turned to be somewhat larger than that found in the present work.

## Methods

**DFT molecular dynamics.** Our DFT molecular dynamics were performed with the VASP code[31]. We used the projector-augmented wave method[31,37] to describe the interactions between the electrons and the ions, and expanded the single-particle orbitals as linear combinations of plane waves (PW), including PW with maximum energies of 293 eV. For the 16-atom bcc SCs, the Brillouin zone was sampled using the 0.25 0.25 0.25 point only (in units of reciprocal lattice vectors) and for the 36-atom HCP SCs using a $3 \times 3 \times 3$ Monckhorst–Pack grid. For larger cells, we used the $\Gamma$ point only. The time step was 1 fs, and the temperature was controlled using a combination of Nosé[38] and Andersen[39] thermostats.

**DFT+DMFT electronic structure and transport calculations.** The DFT+DMFT calculations were performed using a full-potential self-consistent in the charge-density approach[40,41] based on the Wien-2k code[42] and TRIQS library[43,44]. These calculations were carried out for a set of 8 distorted $2 \times 2 \times 2$ bcc supercells (SCs) with the volume 7 Å$^3$ per atom randomly drawn from configurations produced by DFT molecular dynamics simulations. All our DFT+DMFT calculations for the SCs were done for the temperature $T = 5802$ K ($\beta = 1/T = 2$ eV$^{-1}$). We employed the same values of the Coulomb interaction parameters $U = 5.0$ eV, $J_H = 0.93$ eV as in the previous study of the conductivity in perfect hcp[12], and the energy window [−12.2 eV, 4.0 eV] around the Fermi level for the Kohn–Sham states used to construct Wannier orbitals representing Fe 3d states. The DMFT impurity problem was solved by the hybridization–expansion quantum Monte Carlo impurity solver[45]. The density–density approximation was employed for the interaction vertex (which leads to an overestimation of the EES thermal conductivity by 23 and 29% for the perfect hcp and bcc iron phases, see Fig. 1 and ref. [21]), and the nondiagonal elements of the bath Green's function were neglected in the impurity problem.

Each SC was first converged by about 20 fully self-consistent DFT+DMFT iterations with subsequent 10 additional DMFT cycles using the converged Kohn–Sham Hamiltonian. Each Monte Carlo run employed 10$^{10}$ Monte Carlo moves and 200 moves/measurement. Subsequently, 25 additional runs were carried out starting from the same converged value of the DMFT bath Green's function and resetting the random sequence. The 25 self-energies thus calculated were then averaged to obtain the final high-precision DMFT self-energy, which was analytically continued using the Maximum Entropy method[46] to the real-energy axis. Such full DFT+DMFT run for a single 16-atom cell typically took about 7 days of calculation time on a 64-core cluster.

Our DMFT conductivity calculations were carried out within the Kubo linear-response formalism. Namely, the electrical and thermal conductivity reads

$$\sigma_{\alpha\alpha'} = \frac{e^2}{k_B T} K^0_{\alpha\alpha'}, \qquad (4)$$

$$\kappa_{\alpha\alpha'} = k_B \left[ K^2_{\alpha\alpha'} - \frac{(K^1_{\alpha\alpha'})^2}{K^0_{\alpha\alpha'}} \right], \qquad (5)$$

where $\alpha$ is the direction ($x$, $y$, or $z$), $k_B$ is the Boltzmann constant. The kinetic coefficients $K^n_{\alpha\alpha'}$ read

$$K^n_{\alpha\alpha'} = 2\pi\hbar \int d\omega (\beta\omega)^n f(\omega) f(-\omega) \Gamma^{\alpha\alpha'}(\omega, \omega), \qquad (6)$$

where 2 is the spin factor, $f(\omega)$ is the Fermi function, and the transport distribution $\Gamma^{\alpha\alpha'}$ is given by

$$\Gamma^{\alpha\alpha'}(\omega_1, \omega_2) = \frac{1}{V} \sum_{\mathbf{k}} \mathrm{Tr}\left( v^\alpha(\mathbf{k}) A(\mathbf{k}, \omega_1) v^{\alpha'}(\mathbf{k}) A(\mathbf{k}, \omega_2) \right), \qquad (7)$$

where $V$ is the unit-cell volume, $A(\mathbf{k}, \omega)$ is the DMFT spectral function, and $v^\alpha(\mathbf{k})$ is the velocity, see ref. [44]. The optical conductivity is evaluated from the transport distribution as follows:

$$\sigma_{\alpha\alpha'}(\Omega) = 2\pi e^2 \hbar \int d\omega \Gamma^{\alpha\alpha'}(\omega + \Omega/2, \omega - \Omega/2) \frac{f(\omega - \Omega/2) - f(\omega + \Omega/2)}{\Omega}. \qquad (8)$$

Without the additional EES contribution evaluated by DMFT, our approach would reduce to the conventional Kubo–Greenwood DFT formalism routinely used to evaluate the ELS contribution to the resistivity[11,47]. The present approach thus additionally takes the EES into account. In the conductivity calculations for the bcc 16-atom, bcc 54-atom, hcp 36-atom, and hcp 150-atom SCs, we employed $10 \times 10 \times 10$, $5 \times 5 \times 5$, $7 \times 7 \times 6$, and $4 \times 4 \times 4$ **k** mesh in the full Brillouin zone, respectively.

The total thermal conductivity along the [100] direction thus calculated and averaged over all 8 distorted $2 \times 2 \times 2$ SCs is equal to 245 Wm$^{-1}$ K$^{-1}$. However, the 16-atom $2 \times 2 \times 2$ SC is not sufficient for a precise evaluation of the conductivity in bcc-Fe. As shown in Supplementary Fig. 2, one needs at least $3 \times 3 \times 3$ 54-atom bcc SCs to reach the convergence for the ELS conductivity. Full DFT+DMFT calculations are too time-consuming for these 54-atom supercells. Instead, a randomly chosen set of eight $3 \times 3 \times 3$ supercells was drawn from our MD simulations and converged in DFT. The average self-energy $\langle \Sigma(i\omega_n) \rangle$ previously obtained for the $2 \times 2 \times 2$ set (the blue curve in Fig. 2a) was subsequently inserted at each site together with the average double-counting term. Once the chemical potential was found, the conductivity was calculated using $\langle \Sigma(i\omega_n) \rangle$ analytically continued to real frequencies (Fig. 2b). This approach is based on the observation that the conductivity becomes insensitive to the site and orbital dependence of self-energy at the realistic distortion levels, see Fig. 3 and the corresponding discussion. We also benchmarked this procedure on the $2 \times 2 \times 2$ set; thus, calculated conductivities are in a good agreement with the full calculations, with the resulting average $\kappa = 232$ Wm$^{-1}$ K$^{-1}$ compared with 245 Wm$^{-1}$ K$^{-1}$ for full calculation in which the site and orbital dependence is retained. The resulting total thermal conductivity evaluated with $3 \times 3 \times 3$ SCs and shown in Fig. 1, $\kappa = 220$ Wm$^{-1}$ K$^{-1}$, is only slightly reduced compared with the magnitude obtained with the $2 \times 2 \times 2$ SCs (see Supplementary Fig. 1).

Whereas the size of our SCs is insufficient to ensure dynamic stability of the bcc structure—for that a tour de force with 1024 atoms was needed in Belonoshko et al.[18]—the choice of the unit-cell shape constrains the dynamics of ions to those representative of the bcc structure in the thermodynamic limit, especially when the influence on the integrated quantities such as conductivity is considered. Moreover, in the relevant range of high temperatures, the EES is due to on-site electronic correlations that are determined by the local environment of each site. The local environment is unlikely to be highly sensitive to large-scale complex lattice dynamics effects, e.g., the "self-diffusion" mechanism proposed in ref. [18] as the origin of bcc-Fe stability. The increase in SC size, if anything, will be impacting the EES by inducing more random thermal disorder on the local level, since the constrains on atom movements due to the periodic boundary conditions are relaxed. One may notice that the magnitude of the average electronic self-energy in bcc diminishes with increasing thermal disorder (Fig. 2, in particular, notice the suppression of the average bcc SC self-energy with respect to the $e_g$-orbital one of the perfect bcc). Given that in bigger unit cells the motion of atoms is constrained less, the EES is thus likely to be suppressed even further. Hence, our main result of a weak EES should be robust with respect to the SC size.

We applied the same framework to evaluate the conductivity in the hcp phase. To reduce the computational effort in this case, we made use of the fact that the conductivity varies little between different SCs, as one sees in the example of bcc (Fig. 1). Hence, we chose a single representative 36-atom hcp SC ($3 \times 3 \times 2$ in crystallographic unit cells with $c/a = 1.6$ and volume 7 Å$^3$ per atom) from a set randomly drawn from our MD simulations. The value of calculated ELS conductivity for the chosen SC was the closest to the average over this set of 15 SC configurations. For that 36-atom SC, we performed the same full DFT+DMFT

calculation as described above for the set of $2 \times 2 \times 2$ bcc SCs. Parameters of this calculation (the temperature $T = 5802$ K, the values of $U$, $J_H$, the energy window for Wannier construction, and number of quantum Monte Carlo cycles) were the same as for the bcc case. We performed 10 self-consistent DFT+DMFT iterations, with 20 additional runs from the same converged bath Green's function to obtain the high-precision self-energy for the analytical continuation. Transport calculations were performed as described above for the case of bcc. The conductivity values reported below are averaged over 3 directions: $c$, and two in-plane ones, $a$ and $\perp a$. The resulting value for the total thermal conductivity of the 36-atom hcp SC is 234 Wm$^{-1}$ K$^{-1}$. Analogously to the case of bcc, we checked the convergence of our result with respect to the SC size. To that end, we chose a representative 150-atom hcp SC ($5 \times 5 \times 3$ of crystallographic hcp unit cells) from the corresponding MD dynamics simulations. This SC size is sufficient to virtually converge the ELS conductivity with respect to SC size, as we verified by calculating the ELS conductivity for yet bigger hcp SCs. The chosen 150-atom SC had a ELS conductivity close to the average one. Subsequently, we inserted the site- and orbital-averaged self-energy $\langle \Sigma(i\omega_n) \rangle$, obtained for the 36-atom SC (black curves in Fig. 2a and b), into this 150-atom SC and calculated the total conductivity. The resulting values of total electrical and thermal conductivities, which are cited in the "Results" section, are about 9% lower than those for 36-atom SC; this moderate reduction of total conductivity with increasing SC size is similar to that in bcc-Fe as noted above. By inserting the same average self-energy into the 36-atom SC, we verified that the value of conductivity thus calculated differs by only 1% from the one, cited above, evaluated for the same 36-atom SC with the original site- and orbital-dependent self-energy.

By evaluating the ELS contribution for the same 150-atom SC in the same framework (i.e., by setting the self-energy to zero in the spectral functions $A(\mathbf{k}, \omega)$ in Eq. (7)), we also extracted the reduction of total conductivity due to the EES. The ELS contribution calculated in this way, 280 Wm$^{-1}$ K$^{-1}$, closely agreed with that calculated for the same SC by the standard DFT approach of refs. [31,32].

**DFT transport calculations.** For the sake of comparison, the ELS contribution was also computed separately using the DFT-based framework of refs. [11] and [47]. These DFT calculations of the ELS-only electrical and thermal conductivities were performed via the Kubo–Greenwood and the Chester–Thellung–Kubo–Greenwood[48] formula, respectively, as implemented in VASP[32]. We employed settings similar to those employed in ref. [11], by averaging over 72 statistically independent configurations on cells, including between 16 and 250 atoms.

## Data availability
The data that support the findings of this study are available from the corresponding author on reasonable request.

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

## Acknowledgements

We are thankful to I.A. Abrikosov and S.I. Simak for valuable discussions and critical reading of the paper. Useful discussions with G. Sangiovanni, M. Ferrero, and J. Vuči-čević are also acknowledged. L.V.P. acknowledges support of the European Research Council grant ERC-319286-QMAC as well as computational resources of the Swedish National Infrastructure for Computing (SNIC) at the National Supercomputer Centre (NSC). L.V.P. is grateful to the computer team at CPHT for support. J.M. acknowledges support of Program P1-0044 of Slovenian Research Agency. D.A. and M.P. acknowledge support from the Natural Environment Research Council (NERC) Grant No. NE/M000990/1 and No. NE/R000425/1. The DFT calculations are performed on the Monsoon2 system, a collaborative facility supplied under the Joint Weather and Climate Research Programme, a strategic partnership between the UK Met Office and NERC. Calculations are also performed at University College London Research Computing on the Materials and Molecular Modelling hub Grant No. EP/P020194/1 and on the Oak Ridge Leadership Computing Facility (Contract No. DE-AC05-00OR22725).

## Author contributions

L.V.P. carried out the DFT + DMFT electronic structure and transport calculations. D.A. and M.P. carried out the DFT molecular dynamics and transport calculations. J.M. derived the analytical expressions for the simplified model. L.V.P., J.M., and D.A. wrote the paper.

## Competing interests

The authors declare no competing interests.
