## [Peer Review File · Nature Communications]

Reviewers' comments:

Reviewer #1 (Remarks to the Author):

The authors attempt to compute the thermal conductivity of the bcc Fe at the PT conditions of the inner core. The stability of the bcc Fe in the IC was demonstrated by Belonoshko et al. (Nature, 2003) for the EAM model of iron. In 2017, Belonoshko et al. (Nature Geoscience) showed by ab initio MD, that the bcc Fe is stabilized by the mechanism of self-diffusion (the authors call this mechanism 'strong anharmonic vibrations') that allows to release the deviatoric stress. It was demonstrated that to activate that mechanism large (by ab initio MD standards) cells (on the order of 1000 atoms and more to reach the convergence) are needed. Indeed, Godwal et al. (EPSL, 2015) demonstrated that Vocadlo et al. (Nature, 2003) simulations led to configurations of bcc subjected to deviatoric stress, therefore Vocadlo et al. did not demonstrate the mechanical (and the more so thermodynamical) stabilization of the bcc phase.

In the submitted manuscript, the authors again try to study small bcc cells where the mechanism of the bcc stabilization is not and can not be activated (Belonoshko et al., Nature Geoscience, 2017). Since the thermal disordering is critical for correct calculations of the thermal conductivity (Fig. 1) and the 2x2x2 and 3x3x3 cells are too small to capture that disorder correctly, it is highly likely (to put it mildly) that the calculated thermal conductivity is not correct. The authors are trying to estimate the impact of the supercell size on the obtained data by performing calculations for 2x2x2 and 3x3x3 supercells. They obtain the difference of 6 percent. This is not the evidence for the size converged results, because both cells are way too small for the stabilization mechanism to be in place and for the relevant configurations to emerge. The disorder in the large supercells (where the liquid-like diffusion occurs) is much larger than in small ones and that will likely dramatically affect the results.

Since, according to authors, they can not calculate thermal conductivity for large cells, revision will not help. Therefore, I suggest a rejection.

Reviewer #2 (Remarks to the Author):

Electronic correlations and transport in iron at Earth's core conditions

The paper presents the results and analysis of calculations of the electrical and thermal conductivities of bcc iron at Earth's inner core conditions, including the effects of both electron-electron scattering and electron-lattice scattering. Precise values of the thermal conductivity of iron at these conditions are essential for constraining the age and dynamics of the core, with significant implications for magnetic field generation and thermal evolution of Earth. However, experimental measurements of these properties at the relevant conditions are challenging and, results from much lower temperatures and pressure are often extrapolated to predict value at inner core conditions.

Several recent first-principles studies have reported higher values than those predict by the extrapolation measurements made at much lower temperature and pressures. However, these neglected the influence of electron correlation and their validity has been questioned by later studies. Two scattering mechanisms exist: electron-electron scattering and electron-lattice scattering. Previous studies have tended to either neglect or approximate one mechanism, whereas the present study provides a full and detailed consideration of both.

The authors find that electron-lattice scattering dominates, with the inclusion of electron-electron scattering leading to a reduction in the total calculated thermal conductivity of no more than 20 %.

My feeling is that the results are significant; in particular, the finding that the electron-electron scattering is suppressed by thermal disorder. It is quite a technical paper, but comparable to the recent Nature Communications paper by Hausoel et al. (2017) on a similar topic and so I believe it fits within the aims and scope of the journal.

Some comments and questions:

- I am not an expert in DMFT and so will leave assessment of the method to the other reviewers. They authors appear to have carried out convergence tests to investigate possible finite-size effects.
- Is it reasonable to assume same U value for hcp and bcc iron? Were the molecular dynamics calculations performed using LDA+U?
- Comparing the result for bcc iron in Fig 2 in the current manuscript with those for fcc Ni in Supplementary Fig. 6 of Hausoel et al. (2017), can the author suggest why the effect of thermal disorder is important for bcc iron and not for fcc Ni?
- The authors do not consider possible saturation of conductivity reported by Gomi et al. (2013) and included in the calculations of Xu et al. (2018). Can the authors comment on whether they think that saturation occur and if so, its possible influence on their values.
- Though I understand that this is mainly a theoretical study of the importance of the different scattering mechanisms in bcc iron, the motivation for the paper is a better understanding of Earth's core and so I think it would be useful for the authors to comment on
- Is a 20 % reduction in thermal conductivity significant? Will it have a major effect on core dynamics?
- The results presented are for pure bcc iron. What are the authors thoughts on how things will change once impurities, such as Ni, O and S, expected in the core are included.

Some further minor comments:

- P1 Consider replacing 'importantly affected' with 'greatly affected'.
 - P2 Consider replacing 'close to the melting' with 'close to melting'.
 - P2 Consider replacing 'standardly used' with 'routinely used'.
 - P2 Consider replacing 'notice that a spread' with 'notice that the spread'.
- FIG 1. It would be helpful in the caption to say what density Earth's core volume corresponds to.

Reviewer #3 (Remarks to the Author):

The manuscript by Pourovskii et al. reports on the thermal and electrical conductivities of bcc iron at Earth's core pressure. Their calculations take into account electron-electron scattering as well as thermal disorder associated to the high temperatures at the center of the planet.

The long-standing issue that Pourovskii et al. want to address regards the values of these conductivities obtained with density functional theory-based calculations, that are too large to explain the convection forces needed to currently sustain the geodynamo.

The conclusions of the manuscript are that the thermal disorder reduces the conductivities in bcc iron and that the electron-electron scattering does not play a dominant role.

Topic-wise the present manuscript is interesting. Below I list some important points which however in my opinion ought to be considered in order to make it meet the high goals of Nature Communications.

-language/style: overall it does not read like a Nature Communications paper. The style is rather closer to a more specialized or even technical paper. In order to reach the standards of presentation of a Nature paper, the Authors should definitely make an effort and improve many parts of the manuscript.

-poor reference to previous works: the way some references are cited and discussed needs a substantial revision. Here I specifically discuss the example of Ref. 19 which I happen to know well, but many other References may suffer from the same problem. At the end of page 1 a discussion about the van Hove singularity and its slowing down effect on electrons cites only Ref. 22 surprisingly ignoring the fact that this is a much more general mechanism discussed in, e.g. Phys. Met. Metallogr. 76, 247–299 (1993) or for the specific case of hcp iron with Ni impurities in Ref. 19, indeed.

Further, in the two paragraphs at the beginning of page 2, credits to previous works is also missing, in particular when the authors claim that "disorder and many-electron effects has not been directly assessed in previous theoretical studies of transport in considering electron transport in bcc-Fe under conditions relevant to the inner core." This may be strictly true for transport calculations but in Ref. 19 a molecular dynamics + DMFT calculation has been explicitly carried out (even though indeed only showing its effects on the scattering rate) and hence -- at least for the scattering rate -- this study ought to be mentioned here. A similar lack of credits to previous MD+DMFT calculations in a similar context can be found in the second paragraph of the same page.

-conclusions not supported by explicit calculations: the considerations about the possible similarity of results for the hcp phase made at the end of the manuscript are based on mere expectations rather than on results. While speculation might be inspiring in certain cases, I am afraid that they cannot be made on this issue, at least at this stage. As discussed in Ref. 19, in hcp iron the density of states is much more asymmetric and the chemical potential is not right inside a region of high density of states, as in bcc iron. Hence, without an explicit calculations, no reliable "extrapolation" of bcc behavior for corresponding hcp case can be made safely. The authors may either remove this speculative part completely, or discuss in a critical way all possibilities, or of course show further calculations, which however I am perfectly aware that they will be time consuming.

-one further comment: I like the "game" of arbitrary increasing/decreasing $\text{Im}\Sigma$ but are the Kramers-Kronig relations between the real and imaginary part not affected by that? The real part ought to be accordingly changed when α is varied.

Giorgio Sangiovanni

We are thankful to the Referees for carefully reading our manuscript, as well as for their constructive criticism and useful suggestions.

Referee #1 has argued that our calculations cannot properly describe the bcc phase due to the limited size of the unit cell. Whereas this limitation (as we argue in detail below) is actually not a limitation, a direct way to test our general conclusions is to consider also the hcp structure where the issues raised by the referee do not arise at all. We have done these new calculations and reach full agreement with our previous results.

Referee #2 has positively evaluated the manuscript, in particular pointing out the significance of our results for the ongoing debate on the conductivity of Earth's core matter. The questions and comments raised in this report are fully addressed in our reply.

Referee #3 has raised concerns regarding how the previous literature is acknowledged. We corrected this. He also argued: "conclusions not supported by explicit calculations: the considerations about the possible similarity of results for the hcp phase made at the end of the manuscript are based on mere expectations rather than on results. While speculation might be inspiring in certain cases, I am afraid that they cannot be made on this issue, at this stage...". By doing explicit calculations for hcp structure, as mentioned above, we also addressed this point.

Our detailed responses to all Referees' questions and comments are listed below.

Reviewer #1 (Remarks to the Author):

The authors attempt to compute the thermal conductivity of the bcc Fe at the PT conditions of the inner core. The stability of the bcc Fe in the IC was demonstrated by Belonoshko et al. (Nature, 2003) for the EAM model of iron. In 2017, Belonoshko et al. (Nature Geoscience) showed by ab initio MD, that the bcc Fe is stabilized by the mechanism of self-diffusion (the authors call this mechanism 'strong anharmonic vibrations') that allows to release the deviatoric stress. It was demonstrated that to activate that mechanism large (by ab initio MD standards) cells (on the order of 1000 atoms and more to reach the convergence) are needed. Indeed, Godwal et al. (EPSL, 2015) demonstrated that Voadlo et al. (Nature, 2003) simulations led to configurations of bcc subjected to deviatoric stress, therefore Voadlo et al. did not demonstrate the mechanical (and the more so thermodynamical) stabilization of the bcc phase.

In the submitted manuscript, the authors again try to study small bcc cells where the mechanism of the bcc stabilization is not and can not be activated (Belonoshko et al., Nature Geoscience, 2017). Since the thermal disordering is critical for correct calculations of the thermal conductivity (Fig. 1) and the 2x2x2 and 3x3x3 cells are too small to capture that disorder correctly, it is highly likely (to put it mildly) that the calculated thermal conductivity is not correct. The authors are trying to estimate the impact of the supercell size on the obtained data by performing calculations for 2x2x2 and 3x3x3 supercells. They obtain the difference of 6 percent. This is not the evidence for the size converged results, because both cells are way too small for the stabilization mechanism to be in place and for the relevant configurations to emerge. The disorder in the large supercells (where the liquid-like diffusion occurs) is much larger than in small ones and that will likely dramatically affect the results.

Since, according to authors, they can not calculate thermal conductivity for large cells, revision will not help. Therefore, I suggest a rejection.

The fact that we consider the bcc structure is in part motivated by the nice work done by Belonoshko et al., indeed. We are well aware that the structural stability against hcp that was demonstrated in that work only occurs for large unit cells. We base in part our work on their conclusions and assume that the structure is bcc by imposing the corresponding shape of the simulation cell.

Given this assumption, there is no reason to believe that the gross features (such as the density of states) that determine electronic correlations would be severely influenced by the size of the simulation cell. There are no reports of this to be the case in the literature. Furthermore, at the temperature of interest, the density of states for the hcp structure and even for a liquid do not differ a lot from the one of the bcc as we show in Fig. 2d of the revised manuscript. Mechanisms of dynamic and thermodynamic stabilization of bcc are thus subtle, as the free energy of all solid phases at the inner core conditions is also expected to be very similar.

By no means we want to say that our unit cells are sufficiently large to exhibit dynamic structural stability. Instead, we insist that our unit cells are sufficiently large to study the influence of thermal disorder on the electronic correlations and their mutual effects on the electronic transport. Our general finding is that thermal disorder does not increase electronic correlations and that the overall effect of the electronic correlations on transport is much smaller than the one of thermal disorder itself. The dynamical instability of bcc is highly unlikely to affect this conclusion. On this point we would like to quote a private communication from Prof. Sangiovanni (Referee #3):

since electron-electron interaction contributes a finite lifetime, you want to assess whether or not this wins against thermal and electron-lattice effects. Of course if you had used the 4.000 atom unit cell by Belonoshko you would have taken some anharmonicity effects more accurately into account but since you find that already with a small cell (i.e. with more constrained ions' motions) electron-electron interaction does not prevail, it is hard to imagine that in the 4.000 atom unit cell the opposite can occur. The more freedom for thermal disorder present in such a big cell will result in even more room for electron-lattice effects and I would therefore expect them to be more active. On the other hand, the electron-electron part, since at these temperature you can safely neglect non-local correlations generated by local interactions, will not be particularly affected by the size of the unit cell.

To settle the debate, we calculated the electronic transport including thermal disorder also for the case of the hcp structure. The difference between the hcp and bcc lattice structures is expected more significant than the difference between bcc calculated with different simulation cells. The issues related to the dynamical stability do not arise in hcp. As we show, the dynamically unstable bcc and dynamically stable hcp are found to feature almost the same electrical and thermal conductivity. In both the bcc and hcp case electron-electron scattering suppresses the thermal conductivity by about 20%. This confirms our understanding that the magnitude of the electron-electron scattering depends on gross features of DOS only. Our main finding – that the electron-electron scattering is weak and , hence, in geophysical models for iron in Earth's core one should assume high values of conductivity dominated by the electron-lattice scattering – is thus robust.”

Reviewer #2 (Remarks to the Author):

Electronic correlations and transport in iron at Earth's core conditions

The paper presents the results and analysis of calculations of the electrical and thermal conductivities of bcc iron at Earth's inner core conditions, including the effects of both electron-electron scattering and electron-lattice scattering. Precise values of the thermal conductivity of iron at these conditions are essential for constraining the age and dynamics of the core, with significant implications for magnetic field generation and thermal evolution of Earth. However, experimental measurements of these properties at the relevant conditions are challenging and, results from much lower temperatures and pressure are often extrapolated to predict value at inner core conditions.

Several recent first-principles studies have reported higher values than those predict by the extrapolation measurements made at much lower temperature and pressures. However, these neglected the influence of electron correlation and their validity has been questioned by later studies. Two scattering mechanisms exist: electron-electron scattering and electron-lattice scattering. Previous studies have tended to either neglect or approximate one mechanism, whereas the present study provides a full and detailed consideration of both.

The authors find that electron-lattice scattering dominates, with the inclusion of electron-electron scattering leading to a reduction in the total calculated thermal conductivity of no more than 20 %.

My feeling is that the results are significant; in particular, the finding that the electron-electron scattering is suppressed by thermal disorder. It is quite a technical paper, but comparable to the recent Nature Communications paper by Hausoel et al. (2017) on a similar topic and so I believe it fits within the aims and scope of the journal.

We thank the referee for this positive assessment. As any scientific result, it also has technical aspects, but our general finding that can be phrased as “electron-electron repulsion does not matter, in the end” is well intelligible and of interest to a broad community of physicists, geophysicists and other scientists following the debate on the “Transport crisis” in Earth's Core.

Some comments and questions:

- I am not an expert in DMFT and so will leave assessment of the method to the other reviewers. They authors appear to have carried out convergence tests to investigate possible finite-size effects.

The generality of our conclusions is demonstrated by repeating the calculation also for the hcp structure.

- Is it reasonable to assume same U value for hcp and bcc iron? Were the molecular dynamics calculations performed using LDA+U?

Values of U are not known but they are not expected to vary substantially with the crystalline structure: unlike in oxides where the proximity of the oxygen states has strong influence on the electronic screening and hence the value of U depends a lot

on the case, this is not the case presently. Because there is no reason to assume a vary different value of U , we chose the same as in our previous calculations.

Equally importantly, our conclusions do not change if somewhat higher U is assumed. Still the electronic correlations will not be increased by disorder, and even when assuming scattering rates that are unphysically large, still the dominating role in scattering is played by the thermal disorder (our Fig. 4 and the discussion therein). As such, we reassert, our general conclusion that is of interest to the general audience does not depend on the details.

Our molecular dynamics simulations are carried out using DFT with GGA exchange-correlation potential. Iron at high pressure remains a rather itinerant system, for which the use of LDA+ U is not justified. It is unlikely that DMFT corrections to the atomic movement in molecular dynamics would be significant for such an itinerant system; anyhow, MD calculations using DFT+DMFT are prohibitively expensive at present and will remain so in the near future.

- Comparing the result for bcc iron in Fig 2 in the current manuscript with those for fcc Ni in Supplementary Fig. 6 of Hausoel et al. (2017), can the author suggest why the effect of thermal disorder is important for bcc iron and not for fcc Ni?

Overall, the electron-electron scattering in fcc Ni is rather weak, as evidenced by a Fermi-liquid like self-energy (black curve in Supplementary Fig. 6 of Hausoel et al.) and the ratio of the scattering range to temperature $\Gamma/T \ll 1$ also at low T (see Fig. 4c in the same reference). The magnitude of Imaginary part of self-energy in the zero-frequency limit is well below that for the e_g states of the perfect bcc (Fig. 2a of our work). As discussed by Hausoel et al the peculiarities associated to the van-Hove singularity do affect screening of magnetic moment and magnetic properties, but apparently do not lead to a large electron scattering. This might be associated to the fact that the occupancy of Ni is larger and hence is less prone to the Hund's metal physics (that applies in particular to occupancy $N=6,4$ of the d-shell) that makes the electronic correlations more sensitive to the details of the DOS at low energies.

Hence, the behavior of fcc Ni seems to be closer to that of hcp Fe, which is a Fermi liquid for the perfect lattice, than to a very large non-Fermi-liquid scattering one finds for the perfect bcc iron. In the revised Fig. 2a we now compare the average self-energy in thermally disordered hcp with that of perfect hcp-Fe; one observes virtually no difference. Hence, the weak effect of thermal disorder on electronic correlations in hcp Fe seems to be quite similar to the result of Hausoel *et al.* for Ni.

- The authors do not consider possible saturation of conductivity reported by Gomi et al. (2013) and included in the calculations of Xu et al. (2018). Can the authors comment on whether they think that saturation occur and if so, its possible influence on their values.

The saturation of the conductivity actually does occur in our calculations, see inset in revised Fig. 3. Notice that the DMFT calculations are able to describe well both the cases with saturating and nonsaturating resistivity. We now comment on this in the revised manuscript, see subsection IC.

- Though I understand that this is mainly a theoretical study of the importance of the different scattering mechanisms in bcc iron, the motivation for the paper is a better understanding of Earth's core and so I think it would be useful for the authors to comment on

- Is a 20 % reduction in thermal conductivity significant? Will it have a major effect on core dynamics?

Well, 20% change does affect the geophysics models. However, still, the scatter of values between different calculations including just thermal disorder might be of the similar magnitude. Hence, we assert again: our main finding can be summarized as: "Electronic correlations do not matter".

- The results presented are for pure bcc iron. What are the authors thoughts on how things will change once impurities, such as Ni, O and S, expected in the core are included.

The impurities will need to be studied separately, however, intuitively thermal disorder is already a disorder, hence the influence of the impurities should be less pronounced when it becomes strong, hence we expect similar suppression of the relative role of the impurities than what we find for the electronic correlations.

Some further minor comments:

P1 Consider replacing 'importantly affected' with 'greatly affected'.

P2 Consider replacing 'close to the melting' with 'close to melting'.

P2 Consider replacing 'standardly used' with 'routinely used'.

P2 Consider replacing 'notice that a spread' with 'notice that the spread'.

FIG 1. It would be helpful in the caption to say what density Earth's core volume corresponds to.

We thank the referee for these suggestions. All of them have been implemented in the revised version.

Reviewer #3 (Remarks to the Author):

The manuscript by Pourovskii et al. reports on the thermal and electrical conductivities of bcc iron at Earth's core pressure. Their calculations take into account electron-electron scattering as well as thermal disorder associated to the high temperatures at the center of the planet.

The long-standing issue that Pourovskii et al. want to address regards the values of these conductivities obtained with density functional theory-based calculations, that are too large to explain the convection forces needed to currently sustain the geodynamo.

The conclusions of the manuscript are that the thermal disorder reduces the conductivities in bcc iron and that the electron-electron scattering does not play a dominant role.

Topic-wise the present manuscript is interesting. Below I list some important points which however in my opinion ought to be considered in order to make it meet the high goals of Nature Communications.

We thank the referee for the positive feedback as well as for his detailed advice that we appreciate and fully implement.

-language/style: overall it does not read like a Nature Communications paper. The style is rather closer to a more specialized or even technical paper. In order to reach the standards of presentation of a Nature paper, the Authors should definitely make an effort and improve many parts of the manuscript.

We tried to rearrange the article into a less technical form. In particular, we moved technical details of our approach from the Result section into Method. As noticed by Referee B, the present paper is comparable in the level of technicality to the work of Hausoel et al. (Ref. 19); some level of technical details is unavoidable in a work on such a complex subject.

- poor reference to previous works: the way some references are cited and discussed needs a substantial revision. Here I specifically discuss the example of Ref. 19 which I happen to know well, but many other References may suffer from the same problem. At the end of page 1 a discussion about the van Hove singularity and its slowing down effect on electrons cites only Ref. 22 surprisingly ignoring the fact that this is a much more general mechanism discussed in, e.g. Phys. Met. Metallogr. 76, 247–299 (1993) or for the specific case of hcp iron with Ni impurities in Ref. 19, indeed. Further, in the two paragraphs at the beginning of page 2, credits to previous works is also missing, in particular when the authors claim that "disorder and many-electron effects has not been directly assessed in previous theoretical studies of transport in considering electron transport in bcc-Fe under conditions relevant to the inner core." This may be strictly true for transport calculations but in Ref. 19 a molecular dynamics + DMFT calculation has been explicitly carried out (even though indeed only showing its effects on the scattering rate) and hence -- at least for the scattering rate -- this study ought to be mentioned here. A similar lack of credits to previous MD+DMFT calculations in a similar context can be found in the second paragraph of the same page.

We agree with this point of the referee. In the revised paper we take care to give a proper credit to the earlier work. In particular, we cite the references provided by the referee and point out that our approach to thermal disorder has been previously used by Hausoel et al. (Ref. 19) and Xu et al. (Ref.20).

-conclusions not supported by explicit calculations: the considerations about the possible similarity of results for the hcp phase made at the end of the manuscript are based on mere expectations rather than on results. While speculation might be inspiring in certain cases, I am afraid that they cannot be made on this issue, at least at this stage. As discussed in Ref. 19, in hcp iron the density of states is much more asymmetric and the chemical potential is not right inside a region of high density of states, as in bcc iron. Hence, without an explicit calculations, no reliable "extrapolation" of bcc behavior for corresponding hcp case can be made safely. The authors may either remove this speculative part completely, or discuss in a critical way all possibilities, or of course show further calculations, which however I am perfectly aware that they will be time consuming.

We have done the calculation for hcp structure. The presented results fully support our statements, which should not be referred as to "speculations" as they relied on the physical insight that we presented in the paper. However, we agree with the referee that concrete result is even better than the physical insight. Hence, we present the new results along the earlier ones demonstrating explicitly the reliability of our qualitative statements.

-one further comment: I like the "game" of arbitrary increasing/decreasing $Im\Sigma$ but are the Kramers-Kronig relations between the real and imaginary part not affected by that? The real part ought to be accordingly changed when α is varied.

We thank the referee for the positive feedback. The point of increasing $Im \Sigma$ is to simulate the effects of potentially increased values of electron-electron scattering. Indeed, the resulting self-energy does not satisfy the Kramers-Kronig relations. However, this is a 'cartoon' self-energy, which is used to capture some key features of the realistic strong electron-electron scattering limit, but it is not meant to satisfy all exact constraints. We can multiply both the real (up to a constant shift) and imaginary part by the same factor in order to satisfy the Kramers-Kronig relations. However, in this case we also keep the self-energy in the same Fermi liquid regime in the large electron-electron scattering limit (as renormalization grows in such a case, too, and hence $Z Im \Sigma$ remains independent of the scaling factor). This is not the physics we want to consider. We know that in a 'Hund's system' like iron the increasing strength of correlations will result in a crossover to a non-Fermi-liquid behavior, where the lifetime of states relatively far away from the Fermi level becomes short. We believe that our approach of scaling $Im \Sigma$ only captures this regime better, than scaling both Re and $Im \Sigma$ uniformly. In the former case we do have large scattering for high-energy states. In the latter case the result is a heavy-fermion coherent Fermi liquid that we do not expect to occur in iron. The problem with this latter approach is also that just the renormalization grows, but the position of kinks in $Re \Sigma$ remain constant, which is not physical. In particular, it brings the states far away from the Fermi level within the energy window opened up by the Fermi function, which would not occur if the kink would be simultaneously shifting to low energy as would be the case if the interaction parameters were actually increased. We added the corresponding discussion to the text (first paragraph of Sec. ID).

For the referee's information we show above a plot of the thermal conductivity vs. α comparing the result of the both scaling approaches. For $\alpha < 6$ both give similar results. In the large α limit the conductivity is smaller with the uniform scaling. As explained above we believe that this behavior is due to an artificial Fermi-liquid regime imposed in the large scattering limit and is not physical in the case of iron.

REVIEWERS' COMMENTS:

Reviewer #2 (Remarks to the Author):

I am satisfied with the authors' response to my comments and criticisms. However, I think it would be helpful for some of the responses in the rebuttal to be incorporated into the main text. In particular, the responses to the comments regarding whether 20 % was a significant reduction in regards to core dynamics, and the effect of impurities. I think they would answer questions readers of the paper would be asking and it would not take much time to add them. That aside, I am satisfied and am happy to recommend publication of the manuscript.

As far as I can tell, I think that they have provided credible responses to the comments of the other referees. The additional calculations on the hcp phase have helped in this respect.

One really minor comment: in the inset of Figure 3 (left-hand panel) it appears to have '0,5' on the y-axis, when I think it should be '0.5'.

Reviewer #3 (Remarks to the Author):

I was asked by the Editor to address the Authors' reply to my comments as well as to those raised by Reviewer #1.

First, I comment on the Authors' response to my report.

-I had suggested them to improve the readability of the text and I am satisfied by their attempt to meet this. Now the manuscript is written more in accordance with the style of Nature Commun papers.

-My next comment was about the literature and I can confirm that the new version give better credit to previous works.

-New calculations for the hcp phase have been made by the Authors, following my suggestion. The results confirm their previous intuition and make the scientific content of this manuscript more robust and convincing.

-As a last comment I had asked clarifications regarding the "game" with the imaginary part of the self-energy. The Authors have answered in great detail and have amended also the manuscript accordingly. I am therefore satisfied with the response to my report and can recommend the manuscript for publication in Nature Commun.

My point of view about the comments by Reviewer #1, in part already quoted by the Authors in their rebuttal letter, is the following:

It is of course not possible to tell the result of a full calculation including electron-electron scattering for the huge cell treated in the "heroic" molecular dynamics calculation by A. Belonoshko et al. Nature Geoscience (2017).

Yet, the goal of the present manuscript is not an accurate calculation of the thermal conductivity in absolute terms. Some of the authors of the current paper have done this without DMFT in a previous Nature paper and there much larger cells have been used. Anyway, since electron-electron interaction contributes a finite lifetime, the authors want to assess whether or not this wins against thermal and electron-lattice effects. Of course, had they used the 4.000 atom unit cell by Belonoshko they would have taken anharmonicity effects much more accurately into account. However, since they find that already with a small cell (i.e. with more constrained ions' motions) electron-electron interaction does not prevail, it is hard to imagine that in the 4.000 atom unit cell the opposite can occur. The more freedom for thermal disorder present in such a big cell will result in even more room for electron-lattice effects and I would therefore expect them to be more active. On the other hand, the electron-electron part, since at these temperature they can safely neglect non-local correlations generated by local interactions, will not be particularly affected by the size of the unit cell.

In the resubmitted version the authors report new calculations for a cell with hcp structure and find a very similar behaviour as that of the bcc one. I believe that this further corroborates their approach and

interpretation.

Therefore, even if the points by Reviewer #1 are absolutely valid in general and well taken in absolute terms, I judge that within the perimeter of the conclusions that the Authors want to draw in the present study, the study done on the small cells is enough.

We are thankful to both Referees for their positive evaluation of our revised version. Our replies to their comments are listed below.

Reviewer #2 (Remarks to the Author):

I am satisfied with the authors' response to my comments and criticisms. However, I think it would be helpful for some of the responses in the rebuttal to be incorporated into the main text. In particular, the responses to the comments regarding whether 20 % was a significant reduction in regards to core dynamics, and the effect of impurities.

Our comments on both issues based on the previous reply have been now included into the last paragraph of the Discussion section.

One really minor comment: in the inset of Figure 3 (left-hand panel) it appears to have '0,5' on the y-axis, when I think it should be '0.5'.

We thank the Referee for careful reading. Indeed, there was a problem with the tick labels format in Figure 3, it has been corrected in the revised version.

Reviewer #3 (Remarks to the Author):

My point of view about the comments by Reviewer #1, in part already quoted by the Authors in their rebuttal letter, is the following: It is of course not possible to tell the result of a full calculation including electron-electron scattering for the huge cell treated in the "heroic" molecular dynamics calculation by A. Belonoshko et al. *Nature Geoscience* (2017). Yet, the goal of the present manuscript is not an accurate calculation of the thermal conductivity in absolute terms. Some of the authors of the current paper have done this without DMFT in a previous *Nature* paper and there much larger cells have been used. Anyway, since electron-electron interaction contributes a finite lifetime, the authors want to assess whether or not this wins against thermal and electron-lattice effects. Of course, had they used the 4.000 atom unit cell by Belonoshko they would have taken anharmonicity effects much more accurately into account. However, since they find that already with a small cell (i.e. with more constrained ions' motions) electron-electron interaction does not prevail, it is hard to imagine that in the 4.000 atom unit cell the opposite can occur. The more freedom for thermal disorder present in such a big cell will result in even more room for electron-lattice effects and I would therefore expect them to be more active. On the other hand, the electron-electron part, since at these temperature they can safely neglect non-local correlations generated by local interactions, will not be particularly affected by the size of the unit cell.

In the resubmitted version the authors report new calculations for a cell with hcp structure and find a very similar behaviour as that of the bcc one. I believe that this further corroborates their approach and interpretation.

We are thankful to Referee for this insightful comment on the effect of supercell size on our conclusions. We fully agree with his conclusion that the SCs used in our simulations is sufficient to draw robust conclusions on the relative impact of electron-electron scattering on transport in iron. We now present our argumentation on this effect of SC size, which is based in part on this comment, in a new paragraph in Methods.